# An interactive geographic information system to inform optimal locations for healthcare services

Calum Nicholson[1,2,3]*, Mark Hanly[4], David S. Celermajer[1,2,3]

1 Central Clinical School, University of Sydney Faculty of Medicine and Health, Darlington, New South Wales, Australia, 2 Clinical Research Group, Heart Research Institute, Newtown, New South Wales, Australia, 3 Cardiology Department, Royal Prince Alfred Hospital, Camperdown, New South Wales, Australia, 4 Centre for Big Data Research in Health, University of New South Wales, Kensington, New South Wales, Australia

* calum.nicholson@hri.org.au

**Data Availability Statement:** Due to patient privacy concerns, the original patient dataset accessed for this study has not been made available. All other data and code used for data cleaning and analysis, and for application development has been made available on GitHub at

## Abstract

Large health datasets can provide evidence for the equitable allocation of healthcare resources and access to care. Geographic information systems (GIS) can help to present this data in a useful way, aiding in health service delivery. An interactive GIS was developed for the adult congenital heart disease service (ACHD) in New South Wales, Australia to demonstrate its feasibility for health service planning. Datasets describing geographic boundaries, area-level demographics, hospital driving times, and the current ACHD patient population were collected, linked, and displayed in an interactive clinic planning tool. The current ACHD service locations were mapped, and tools to compare current and potential locations were provided. Three locations for new clinics in rural areas were selected to demonstrate the application. Introducing new clinics changed the number of rural patients within a 1-hour drive of their nearest clinic from 44·38% to 55.07% (79 patients) and reduced the average driving time from rural areas to the nearest clinic from 2·4 hours to 1·8 hours. The longest driving time was changed from 10·9 hours to 8·9 hours. A de-identified public version of the GIS clinic planning tool is deployed at https://cbdrh.shinyapps.io/ACHD_Dashboard/. This application demonstrates how a freely available and interactive GIS can be used to aid in health service planning. In the context of ACHD, GIS research has shown that adherence to best practice care is impacted by patients' accessibility to specialist services. This project builds on this research by providing opensource tools to build more accessible healthcare services.

## Author summary

It is important that our health system can utilise available data when determining where to place healthcare resources. Spatial data relates to where something is in the world, where a patient lives or where a hospital is located. Geographic information systems can draw insights from this data. Previous research over the past decade has demonstrated

https://github.com/CBDRH/achd-dissertation.git.
The public version of the application presented
here is available at https://cbdrh.shinyapps.io/
ACHD_Dashboard/.

**Funding:** The authors received no specific funding
for this work.

**Competing interests:** I have read the journal's
policy and the authors of the manuscript have the
following competing interests: DC and CN are part
of the Congenital Heart Alliance of Australia and
New Zealand, whose currently work involves the
development of a Congenital Heart Disease
Registry for Australia and New Zealand. Whilst the
current study was not part of this affiliation, future
development of the methodologies described here
will likely make use of this resource.

how these techniques can measure and describe healthcare accessibility. However, it still
requires a certain level of technical skill to use these data. We aimed to demonstrate the
feasibility of developing an interactive application that makes these data accessible to lay
audiences. Using adult congenital heart disease care in New South Wales as an example,
we developed a clinic planning tool that allows users to compare patient accessibility
under hypothetical new clinic locations. As an example, we showed that the addition of
three clinics would increase the number of rural patients within a 1-hour drive of their
nearest clinic from 44% to 55% (79 patients). Our study demonstrates how freely available
data and software can be used to develop tools to inform healthcare planning and resource
allocation.

## Introduction

Equity of access to health care is a critical issue for patients, families, health care providers, and
governments. Health service planners have access to increasingly large amounts of information
to guide the allocation of health care resources. To develop a modern, functional health system,
new tools that aid in the communication and visualisation of health and population data are
required to enable these data to inform policy and improve health care.

The social determinants of health are often used when describing inequity in health care
delivery and outcomes. However, including a spatial perspective to these sociodemographic
factors can develop a more nuanced and practical model of healthcare accessibility [1,2].
Shukla, N et al. [3] provide a review of healthcare accessibility models in Australia and identify
characteristics of healthcare accessibility as spatial (availability and accessibility) and non-spa-
tial (affordability, acceptability, and accommodation). The relationship between distance to
healthcare facilities and healthcare outcomes, the distance-decay association, has been previ-
ously demonstrated [4]. The distance between patients and services is a useful method to mea-
sure spatial aspects of healthcare accessibility.

Non-spatial aspects of healthcare accessibility require an assessment of the many factors
that make up the social determinants of health. The Socio-Economic Index for Areas (SEIFA)
provided by the Australian Bureau of Statistics (ABS) provides four different measures of peo-
ple's ability to participate in society and the resources that they can access [5]. These summary
measures are a useful way to measure disadvantage in the Australia context and can be com-
bined with spatial information to create a comprehensive model of healthcare accessibility. It
is also important for any assessment of healthcare accessibility in Australia to address the well
documented barriers to healthcare that are faced by The Aboriginal and Torres Strait Islander
population [6].

Measurement of healthcare access must consider both the geographic distribution of a
patient population and their healthcare services as well as the social determinants of health
that influence healthcare engagement. This is especially true in jurisdictions like New South
Wales (NSW), which are geographically vast with many patients living in remote areas, far
from urban centres.

Geographic information systems (GIS) are software tools for the collection, analysis, and
visualisation of spatial data and they can be powerful tools for the assessment of healthcare
accessibility. The combination of spatial and non-spatial data can uncover locations with inad-
equate access to health care and inform health resource allocation. This previous research pro-
vides insights by analysing the current state of health services to identify deficiencies and
inequities [7–9]. Kamil, W et al. [10] use a similar model of accessibility in the Australian

context, by making use of the SEIFA measures provided by the ABS to combine both spatial and socio-demographic data. Further research has highlighted the utility of interactive platforms for this field. This includes applications that use GIS to explore epidemiological research, prediction of infectious disease incidence [11] or to measure the geographic distribution of health outcomes [12]. Further applications have provided interactive interfaces that can model of current healthcare accessibility to identify potential gaps [13,14].

Freely available packages in the R Statistical Software, notably *Shiny* for creating interactive web applications [15] and *Leaflet* for displaying data over interactive maps [16], enable the creation of a opensource clinic planning tool that overlays geospatial and sociodemographic data. By providing an accessible and interactive interface to existing datasets, this tool can be used support health service planning at minimal cost.

These concepts were applied to adult congenital heart disease (ACHD) services in NSW, Australia. ACHD is a growing disease population who's increased prevalence requires increasing healthcare resources [17–19]. With complex care needs, all but the simplest congenital cardiac defects should be managed by specially trained ACHD cardiologists [20–26]. Current research highlights the psychological and social impacts of CHD, describing high rates of anxiety and depression, high healthcare utilisation, significant professional disruption, and long travel times for rural CHD patients [27–29].

Given the specialist care needs and high burden of disease of ACHD patients, coupled with the large geographic area and dispersed rural population of NSW, geographic analysis of the current health care services can provide an opportunity to improve health care outcomes. Guidelines specific to the Australian context have suggested that a state-wide ACHD service should include a comprehensive ACHD centre that is supported by smaller, regional services [30]. Under this model, "clinics" are medical services that are attended by an ACHD-trained cardiologist. Permanent and comprehensive services in major urban centres (i.e Sydney) support satellite clinics in rural hospitals where Sydney-based, ACHD-trained cardiologists will travel to support local patient populations. The current project aimed to demonstrate the feasibility of using an interactive GIS to help inform the optimal location for these satellite healthcare services.

## Materials and methods

### Ethics statement

Human Research Ethics Approval was obtained from the Sydney Local Health District Ethics Review Committee (RPAH Zone) (EC00113). The number for this ethics protocol is 2020/ETH00001. Due to the large number of patients involved and the low-risk nature of the research, a waiver of consent was granted by the Human Research Ethics Committee to collect retrospective patient data.

### Analysis software

All analyses were performed using R Statistical Software (v4.0.5) [31]. Data cleaning and manipulation was conducted using the *tidyverse* suite of packages (v1.3.1) [32]. The Clinic planning tool was developed using *shiny* (v1.7.1) [15] and mapping functionality added with *leaflet* (v2.0.4.1) [16]. Base map data is copyrighted to OpenStreetMap contributors under the Open Data Commons Open Database License and available from https://www.openstreetmap.org. Shape files are copyrighted to the Australian Bureau of Statistics provided under a Creative Commons Attribution 4.0 International license and available from https://www.abs.gov.au/AUSSTATS/abs@.nsf/DetailsPage/1270.0.55.001July%202016?OpenDocument#Data. All source code for data preparation and app development is available at: https://github.com/CBDRH/achd-dissertation.git.

## Identification of data sources

The clinic planning tool combines opensource information on community boundaries, socio-demographic data, and hospital driving times with ACHD patient information. The Australian Bureau of Statistics (ABS) provides the Australian Geographical Statistical Standard (ASGS) framework, a standardised and nested set of geographical boundaries called Statistical Areas (SA). The smallest unit, a mesh block, is used to build up increasingly larger statistical areas, from Statistical Area (SA) 1 to SA4. These boundaries provide a standardised method for "the publication of statistics that are comparable and spatially integrated" [33]. This project uses the SA2 level, which has an average population of 10,000 persons and represents communities that are socially and economically connected.

The AGSG framework is linked to census data, which was aggregated to the SA2 area level and downloaded using the Census Table Builder tool provided by the ABS [34]. Two key summary measures from the 2016 Australian Census were integrated into this application. The Index of Relative Socio-economic Disadvantage (IRSD) aggregates key measures from the 2016 census into a score between one and ten, where lower numbers represent higher disadvantage [5], Accessibility and Remoteness Index of Australia (ARIA) is a measure of geographic accessibility, assessing an area's access to service centres classified into five categories: Major Cities, Inner Regional, Outer Regional, Remote, and Very Remote [35]. To enable users to address barriers to healthcare access for Aboriginal or Torres Strait Islander people, their percentages of the total population in each SA2 area was included.

Barbieri and Jorm [36] developed an open access dataset that details the driving time between each SA2 area and each hospital in Australia. This provides a ready-made, network-based measure to assess driving times to hospitals that is integrated with the ASGS framework.

The Adult Congenital Heart Disease Database (ACHD Database) has been managed as a resource by the ACHD service at Royal Prince Alfred Hospital since 2010 [37,38]. It contains information at both the patient level and encounter level. Key measures taken from this dataset were patient's location, diagnosis, disease severity, and clinic attendances.

## Selection and standardisation of patient data

The patient dataset was selected from the ACHD Database. The study period was selected as 01 Jan 2000 to 01 Jul 2022, with the end date being where the data was most up to date at the time of data extraction. Patients who were under the age of 18 at the end of the study period were also excluded as this project aimed to focus on adult patients. Finally, patients with missing address information or outside of NSW or the Australian Capital Territory (ACT) were excluded. 301 patient addresses were manually cleaned to reduce loss of data, this cleaning process was documented and recorded in the GitHub repository. Fig 1 outlines the patients that were excluded.

The boundary data, driving distance measure, and socio-demographic information were provided at the SA2 area level and only required filtering to NSW and ACT boundaries for use in the clinic planning tool. Patient locations are provided as addresses in the ACHD Database and were standardised to the ASGS. Lookup tables are provided by the ABS allow the linking of postcode/suburb combinations to SA2 areas. These lookup tables were used to aggregate patients into SA2 areas for geographic visualisation.

To allow for further investigation of the CHD-specific healthcare requirements that a location might require, diagnosis and disease severity information were included as well. The patient dataset categorises CHD diagnoses using an 'in-house' system which was matched to the European Paediatric Cardiology Code—Short List to allow for a standardised presentation of CHD diagnoses. The methods of diagnosis standardisation and reasons for using this

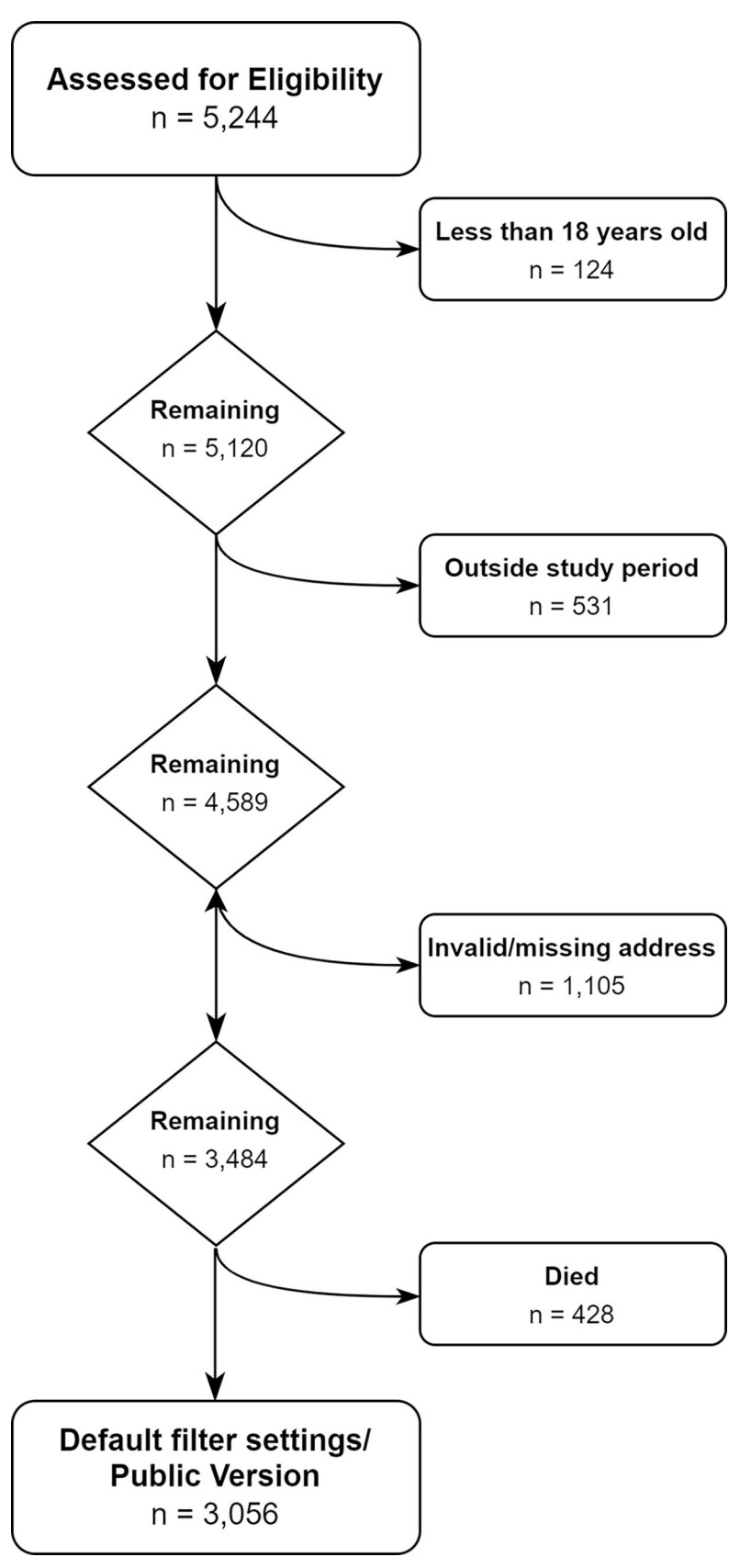

**Fig 1. Patient Selection Flowchart.** The filtering process for patients that were selected for inclusion in the clinic planning too.

standard have been previously described [39]. This patient dataset also provides a clinician-defined measure of anatomic complexity which follows the 2008 American Heart Association ACHD Guidelines [23] and categorises patients as Simple, Moderate, Complex, or Unknown. These two measures were aggregated into SA2 area levels based on patient locations, described above.

To describe clinic attendance of the patient population, dates of attendance at ACHD Clinics were taken for each patient. The time since last clinic attendance was calculated as the difference in time between the most recent date of attendance and the end of the study period (or the date of death for patients that had passed away). Patients with a gap in care longer that three years were determined to be lost to follow up.

## Maintaining patient privacy

Presentation of geographic information for a patient population requires the use and visualisation of identifiable patient information. To ensure that the presentation of this information does not infringe patient privacy whilst also allowing for the presentation and publication of this data, two branches of the clinic planning tool were developed. A private version requires access to offline patient data and allows for full access to patient-level information. A public version includes aggregate data only and masks any geographic patient information below five individuals.

## Results

### Patient demographics

The total number of patients in the ACHD Database at the time of extraction was 5,244 and after filtering for age, valid addresses, and an encounter within the study period, 3,484 patients remained eligible for inclusion in this study and of these 3,056 were still alive (Fig 1).

Of the patients that were alive, the average age of patients was 43·70 years (± 16·65 years) and 1504 (49·21%) of patients were female. On average patients had 3·35 clinic attendances (SD = 2·29 attendances). Their last clinic attendance occurring an average of 4·79 years (SD = 5·42 years) before the end of the study period. The most common CHD diagnoses in this dataset were Bicuspid Aortic Valve (20·98%), Ventricular Septal Defect (20·12%), Atrial Septal Defect (12·08%), Tetralogy of Fallot (11·19%) and Patent Foramen Ovale (10·04%). There were 1,230 patients with Simple CHD, 894 with Moderate CHD, and 456 with Complex CHD. 476 patients had an unknown CHD classification. 41·33% of patients were lost to follow up and 68·13%, 34·00%, and 21·71% of Simple, Moderate, and Complex severity patients were lost to follow up respectively (Table 1).

In all NSW, 49·3% of people were female and 2·3% of people were Aboriginal or Torres Strait Islander. Based on the ARIA classification, 74·45% of the NSW population lives in major cities, 19·02% lives in inner regional areas, 6·03% lives in outer regional areas, 0·42% lives in remote areas and 0·08% lives in very remote areas.

### The clinic planning tool

A de-identified public version of the GIS clinic planning tool is deployed at https://cbdrh.shinyapps.io/ACHD_Dashboard/. Fig 2 provides a screenshot of the clinic planning tool,

**Table 1. Demographic and diagnosis summary of the study population, separated by disease severity.**

| Demographics | All | Simple | Moderate | Complex | Unknown |
|---|---|---|---|---|---|
| N | 3056 | 1230 | 894 | 456 | 476 |
| Age, years (mean (SD)) | 43.70 (16.65) | 51.04 (18.00) | 40.77 (13.59) | 37.01 (10.07) | 36.67 (15.97) |
| Female (N (%)) | 1504 (49.21) | 599 (48.70) | 437 (48.88) | 205 (44.96) | 263 (55.25) |
| Clinic Attendances (mean (SD)) | 3.35 (2.80) | 2.29 (2.01) | 4.27 (2.77) | 5.86 (3.42) | 1.9 (1.19) |
| Years since last clinic attendance, (mean (SD)) | 4.79 (5.42) | 7.82 (5.83) | 3.70 (4.71) | 2.38 (3.35) | 1.31 (2.16) |
| Lost to Follow up (%) | 41.33 | 68.13 | 34.00 | 21.71 | 4.62 |
| **Diagnoses, N (%)** | | | | | |
| Bicuspid Aortic Valve | 641 (20.98) | 432 (35.12) | 155 (17.33) | 6 (1.32) | 48 (10.08) |
| Ventricular Septal Defect | 615 (20.12) | 185 (15.04) | 124 (13.87) | 218 (47.81) | 88 (18.45) |
| Atrial Septal Defect | 545 (17.83) | 297 (24.15) | 108 (12.08) | 65 (14.25) | 75 (15.75) |
| Tetralogy of Fallot | 342 (11.19) | 1 (00.08) | 275 (30.76) | 28 (6.14) | 38 (7.98) |
| Patent Foramen Ovale | 307 (10.04) | 187 (15.20) | 35 (3.91) | 16 (3.51) | 69 (1.45) |

outlining the key sections discussed below; 'Global Filters', 'Map Display', 'Map Customisation', 'Clinic Tables', and 'New Clinic Selection'.

The 'Global Filters' allow a selection of patient data to be made based on key patient demographics, age, sex, mortality, disease severity, period for of clinic attendances, and the time since last clinic visit.

The 'Map Display' can be edited in 'Map Customisation', with filters to display SA2 areas within the selected regions ("Greater Sydney", "Australian Capital Territory", and "Rest of NSW"). Two different area overlays can be selected to change the choropleth on the map. The total 'ACHD population' selection will display the number of ACHD patients in each SA2 area. The 'Driving Time to Nearest Clinic' selection will display the driving time from each SA2 area to the nearest ACHD clinic in NSW (Fig 3).

The clinic planning tool provides an interactive interface to explore the accessibility of current ACHD clinic locations and investigate how new locations change accessibility. The current locations for ACHD care in NSW have been provided in the map as the baseline of health care accessibility. There are two tables in the 'Clinic Tables' section. The 'Current Clinics' table provides details about the current ACHD clinics: number of patients within a 1-hour drive ('Patients'); and the number of patients lost to follow up within a one-hour drive ('ltf'). The 'New Clinics' table shows these details for clinics selected by the user. New locations can be selected in the 'New Clinic Selection' area, which displays existing hospitals where clinics could potentially be located. Any number of hospitals can be sequentially added to the 'New Clinics' Table. Updating the map with new clinics selected will redraw the choropleths and recalculate summary values to consider how the new clinics have affected patient reach and driving times.

Clicking on any SA2 area in the map provides an 'Area Focus' to pop up over the map with details about the ACHD population, their diagnoses, and key socio-demographic characteristics in the selected areas. The user can select as many areas as desired, with each new selection added to the report.

A downloadable report has been included to provide reproducibility in a user's interaction with the application. This report captures the current state of the application's filters together with the corresponding output and combines this information in a downloadable file. The report pulls data from the application into five sections: Patient Data, Area Data, Current ACHD Clinic, New ACHD Clinics, and Area Focus.

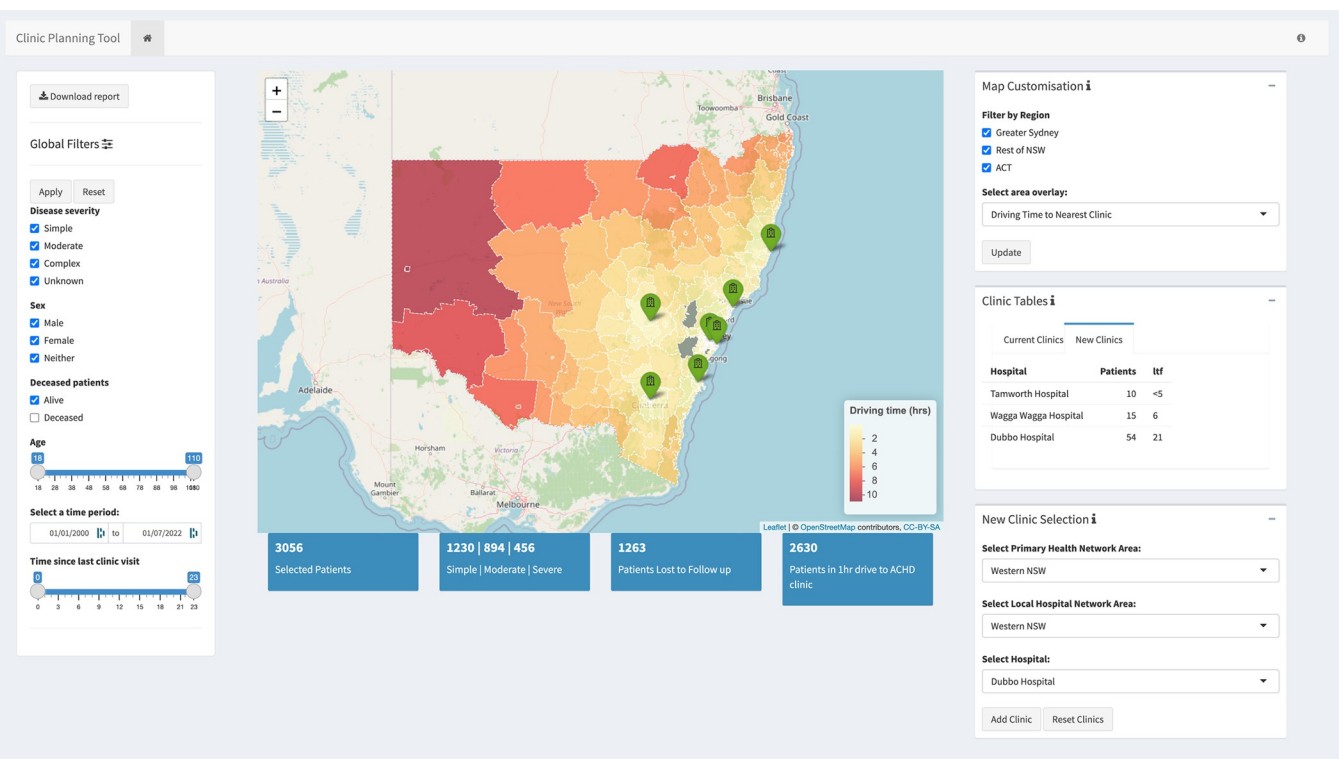

**Fig 2. The Clinic Planning Tool.** A screenshot showing the layout of the clinic planning tool. The Global filters are on the side bar on the left and the Map Customisation, Clinic Tables and New Clinic Selection sections are on the right. The Map is displayed in the centre with key values displayed underneath. Base map data is copyrighted to OpenStreetMap contributors under the Open Data Commons Open Database License and available from https://www.openstreetmap.org. Shape files are copyrighted to the Australian Bureau of Statistics provided under a Creative Commons Attribution 4.0 International license and available from https://www.abs.gov.au/AUSSTATS/abs@.nsf/DetailsPage/1270.0.55.001July%202016?OpenDocument#Data.

## Area demographics around selected hospital

Three hospitals were selected to demonstrate examples of new clinics locations in NSW: Tamworth Hospital, Wagga Wagga Hospital, and Dubbo Hospital. SA2 areas surrounding the three hospitals that were selected as potential new clinic locations (Fig 4).

Four SA2 areas were selected to cover the area around Tamworth Hospital; 'Tamworth—West', 'Tamworth—East', 'Tamworth—North', and 'Tamworth Region'. These areas had IRSD scores of one (highest disadvantage), three, four, and five respectively. 85% of this population was from the Inner Regional remoteness category and 15% Outer Regional remoteness category. 10% of this population was Aboriginal and Torres Strait Islander. Ten patients were in these regions.

Four SA2 areas were selected around Dubbo Hospital; 'Dubbo—East', 'Dubbo—South', 'Dubbo—West', and 'Dubbo Region'. These areas had IRSD scores of three, four, three, and seven respectively. 77% of this population was from the Inner Regional remoteness category and 23% Outer Regional remoteness category. 14% of this population was Aboriginal and Torres Strait Islander. 42 patients were in these regions.

Five SA2 areas were selected around Wagga Wagga Hospital; 'Wagga Wagga—East', 'Wagga Wagga—North', 'Wagga Wagga—South', 'Wagga Wagga—West', and 'Wagga Wagga Region'. These areas had IRSD scores of six, eight, four, three, and six respectively. 94% of this population was from the Inner Regional remoteness category and 6% Outer Regional remoteness category. 5% of this population was Aboriginal and Torres Strait Islander. 14 patients were in these regions.

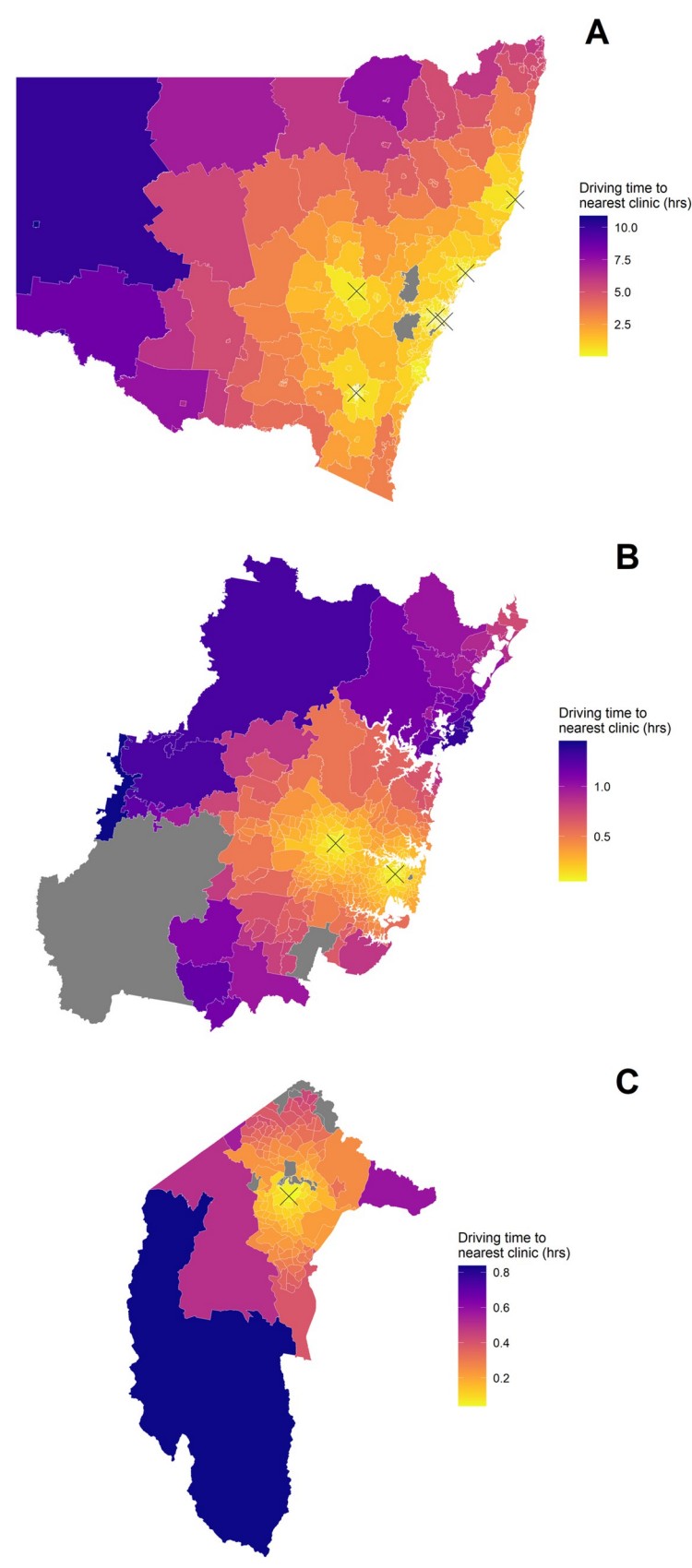

**Fig 3. The driving time from each SA2 area to the nearest ACHD clinic.** (A) New South Wales. (B) Greater Sydney. (C) Australian Capital Territory. "X" marks locations of ACHD Clinics. Areas in grey do not have driving time data provided. Shape files are copyrighted to the Australian Bureau of Statistics provided under a Creative Commons Attribution 4.0 International license and available from https://www.abs.gov.au/AUSSTATS/abs@.nsf/DetailsPage/1270.0.55.001July%202016?OpenDocument#Data.

## Accessibility to ACHD clinics

Using the same three hospital as new clinic locations, changes to patient reach and clinic accessibility were described by the tool.

Of the patients that were currently alive, 83·48% of patients were less than 1-hour drive from a current ACHD clinic, increasing to 86·06% with the new clinic locations included, a total of 79 patients. The average driving time to a current ACHD clinic was 1·13 hours and the longest drive was 10·87 hours. With the three new clinic locations included, the average drive was 0·91 hours and the longest drive was 8·93 hours.

In Greater Sydney, 95·69% of patients were less than 1-hour drive from a current ACHD clinic, the average driving time to a current ACHD clinic was 0·44 hours and the longest drive was 1·45 hours. In the Australian Capital Territory, 100% of patients were less than 1-hour

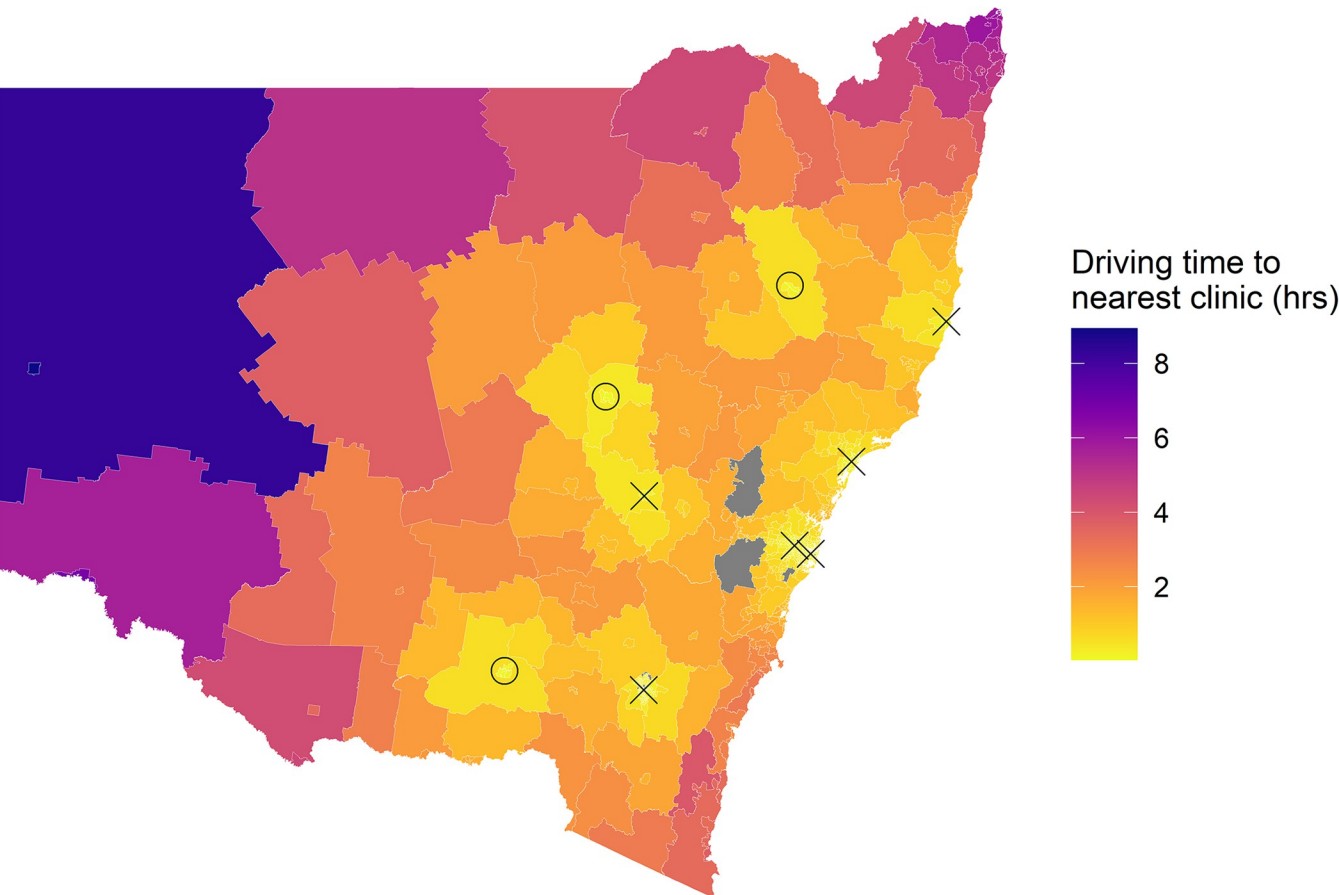

**Fig 4. The driving time from each SA2 area to the nearest ACHD clinic, with proposed changes included.** "X" marks the location of current ACHD clinics. "O" marks the location of proposed ACHD clinics. Areas in gray do not have driving time data provided. Shape files are copyrighted to the Australian Bureau of Statistics provided under a Creative Commons Attribution 4.0 International license and available from https://www.abs.gov.au/AUSSTATS/abs@.nsf/DetailsPage/1270.0.55.001July%202016?OpenDocument#Data.

drive from a current ACHD clinic, the average driving time to a current ACHD clinic was 0·26 hours and the longest drive was 0·84 hours. These figures were unchanged with the new regional clinics included.

In the rest of NSW, 44·38% of patients were less than 1-hour drive from a current ACHD clinic, increasing to 55·07% with the new clinic locations included. The average driving time to a current ACHD clinic was 2·34 hours and the longest drive was 10·87 hours. With the three new clinic locations included, the average drive was 1·76 hours and the longest drive was 8·93 hours.

## Discussion

The ACHD service in NSW provides a useful example of the application of an interactive GIS clinic planning tool to help inform new locations for health services. ACHD patients require specialist care, and these services are not always uniformly available. Accessibility issues are exacerbated by the geographic challenges of NSW. With a large area and a disparate rural population requiring access to care, achieving an equitable distribution of limited health resources is a significant challenge. We have demonstrated that the clinic planning tool can be developed with freely available data and software to provide evidence for healthcare locations. This model can be generalised to other locations and patient populations, and significant benefit may be found when applying these techniques to chronic and complex disease populations, where equitable access is hard to achieve.

Geographic information systems have been demonstrated as useful tools to assess healthcare accessibility. The literature demonstrates how these techniques can assess gaps in healthcare accessibility in both static analyses [7–10] and interactive interfaces [11–14]. The Clinic Planning tool described here builds on this body of work to provide an interactive interface to compare a current state of healthcare locations with a potential future state, providing evidence for the inclusion of healthcare services at new locations. This methodology of applying new locations presents an open-source and reproducible GIS framework for future services location planning.

The clinic planning tool uses both spatial and non-spatial information to assess healthcare accessibility. Non-spatial measures aim to provide information about populations that might face barriers related to the affordability and acceptability of health services. These measures describe the disadvantage, remoteness, and the Aboriginal and Torres Strait Islander population of areas in NSW. These allow the demographics of an area, not just its distance from health services, to inform optimal locations for regional healthcare.

Selecting an appropriate distance measure is important to accurately represent the spatial aspects of this healthcare accessibility model [40]. Both Euclidian and Manhattan distances lack accuracy in less populated and areas and do not accurately reflect transport infrastructure used in the world. Network distance, which uses road networks and a selected mode of transport to measure travel time, can offer a more accurate representation of accessibility to a clinic. As areas become less urbanised, public transport network measures correlate more poorly with network measures such as driving time [41]. The appropriateness of a distance measure relies on the geographic space being measured and the population it contains. This clinic planning tool makes use of a freely available dataset of driving times to measure the ACHD population's access to specialist services [36]. This nation-wide dataset provides an accurate network measure that is tied to standardised geographic units and is an invaluable resource for further spatial epidemiology in Australia.

Using freely available data and software, we demonstrate the feasibility of developing a digital health solution to assist in the assessment and planning of new healthcare locations. We have added to this body of research by demonstrating a simple application that can provide a

link between the GIS researchers who have highlighted these issues and service planners who might provide a solution.

The relationship between demographic and geographic factors and access to specialist CHD services has been well defined in a surgical context. Patients who have more severe CHD, private health insurance, and shorter driving times are more likely to present to specialist cardiac surgery centres [42] and greater distances from CHD centres results in CHD patients having surgery at non-specialist clinics [43] Of those who do attend specialist CHD surgical centres, living further away is associated with a longer length of stay, more complications, and higher cost [44].

Whilst efforts have been made to describe the reach and capacity of current CHD services (both paediatric and adult), [45–47] little has been done to translate these findings into changes in healthcare services. This clinic planning tool allows for this to be determined in NSW, although scaling this to the whole of Australia would benefit future resource planning. The current development of a national registry for CHD patients in Australia and New Zealand will provide a resource to expand the current project to include both countries [48–50].

### Limitations and future directions

Being restricted to NSW, this version of the application assumes that people do not travel across borders to receive healthcare. Given the size and remoteness of NSW, travel into both Queensland and Victoria for CHD care would be a common occurrence. A federal version of this application would address the limitation of the NSW border that is currently present.

Much of the current research utilises outcome data from administrative sources to assessing the relationship between geographic factors and access to CHD care, such as healthcare utilisation, mortality, and cost data. Linking data such as this to the clinic planning tool would provide a great benefit to its utility and is a key area of future development. The current application relies on the number of ACHD patients in an area and the driving time to the nearest clinic to determine the best location for a new clinic. Being able to correlate these data to locations with high rates of hospitalisations, emergency department presentations, primary health care utilisation, mortality, and healthcare costs would allow for key outcomes to be considered when planning for future allocation of health resources for ACHD care. Benderly, M *et. al.* [51] assessed healthcare utilisation of ACHD patients in Israel and demonstrated that this population had higher rates of healthcare utilisation than the general population, with those on geographic peripheries being associated with fewer specialist attendances. Linkage with administrative data has also shown that specialised CHD care is associated with lower medical costs and better economic outcomes for patients [52]. Inclusion of insights drawn from linked administrative data in a clinic planning tool such as the one presented here would allow ACHD services to be delivered with a greater level of precision.

### Conclusion

In this research we demonstrated the feasibility of using an interactive geographic information system to inform optimal locations for regional healthcare, using ACHD care in NSW Australia as an example. The clinic planning tool was developed using open access data and software with minimal resources. Demonstrating the potential for this technology and these techniques to provide real world impact for health care delivery if properly scaled and resourced.

### Acknowledgments

This work was completed through a collaboration between the Centre for Big Data Research in Health at the University of New South Wales and the Adult Congenital Heart Disease service at Royal Prince Alfred Hospital, New South Wales, Australia.

## Author Contributions

**Conceptualization:** Calum Nicholson, David S. Celermajer.

**Data curation:** Calum Nicholson, Mark Hanly.

**Formal analysis:** Calum Nicholson, Mark Hanly.

**Investigation:** Calum Nicholson, Mark Hanly.

**Methodology:** Calum Nicholson, Mark Hanly, David S. Celermajer.

**Project administration:** Calum Nicholson, David S. Celermajer.

**Resources:** Mark Hanly, David S. Celermajer.

**Software:** Calum Nicholson, Mark Hanly.

**Supervision:** Mark Hanly, David S. Celermajer.

**Validation:** Calum Nicholson, Mark Hanly, David S. Celermajer.

**Visualization:** Calum Nicholson, Mark Hanly.

**Writing – original draft:** Calum Nicholson.

**Writing – review & editing:** Mark Hanly, David S. Celermajer.

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
