## [Decision Letter · Decision Letter 0]

1 Dec 2022

PDIG-D-22-00277

A Novel Interactive Geographic Information System to Optimise Health Services Delivery

PLOS Digital Health

Dear Dr. Nicholson,

Thank you for submitting your manuscript to PLOS Digital Health. After careful consideration, we feel that it has merit but does not fully meet PLOS Digital Health's publication criteria as it currently stands. Therefore, we invite you to submit a revised version of the manuscript that addresses the points raised during the review process.

Please submit your revised manuscript within 60 days Jan 30 2023 11:59PM. If you will need more time than this to complete your revisions, please reply to this message or contact the journal office at digitalhealth@plos.org. Please include the following items when submitting your revised manuscript:

We look forward to receiving your revised manuscript.

Kind regards,

Yuan Lai, Ph.D.

Academic Editor

PLOS Digital Health

Journal Requirements:

1. Please amend your online Financial Disclosure statement. If you did not receive any funding for this study, please simply state: “The authors received no specific funding for this work.”

2. We ask that a manuscript source file is provided at Revision. Please upload your manuscript file as a .doc, .docx, or .rtf.

3. Please provide separate figure files in .tif or .eps format only and remove any figures embedded in your manuscript file. Please also ensure that all files are under our size limit of 10MB.

For more information about how to convert your figure files please see our guidelines: https://journals.plos.org/digitalhealth/s/figures

4. Some material included in your submission may be copyrighted. According to PLOS’s copyright policy, authors who use figures or other material (e.g., graphics, clipart, maps) from another author or copyright holder must demonstrate or obtain permission to publish this material under the Creative Commons Attribution 4.0 International (CC BY 4.0) License used by PLOS journals. Please closely review the details of PLOS’s copyright requirements here: PLOS Licenses and Copyright. If you need to request permissions from a copyright holder, you may use PLOS's Copyright Content Permission form.

Potential Copyright Issues:

Figures 2, 3 and 4: please (a) provide a direct link to the base layer of the map (i.e., the country or region border shape) and ensure this is also included in the figure legend; and (b) provide a link to the terms of use / license information for the base layer image or shapefile. We cannot publish proprietary or copyrighted maps (e.g. Google Maps, Mapquest) and the terms of use for your map base layer must be compatible with our CC-BY 4.0 license. 

Additional Editor Comments (if provided):

Reviewers' comments:

Reviewer's Responses to Questions

**Comments to the Author**

1. Does this manuscript meet PLOS Digital Health’s publication criteria? Is the manuscript technically sound, and do the data support the conclusions? The manuscript must describe methodologically and ethically rigorous research with conclusions that are appropriately drawn based on the data presented.

Reviewer #1: Yes

Reviewer #2: Partly

Reviewer #3: Yes

2. Has the statistical analysis been performed appropriately and rigorously?

Reviewer #1: No

Reviewer #2: Yes

Reviewer #3: Yes

3. Have the authors made all data underlying the findings in their manuscript fully available (please refer to the Data Availability Statement at the start of the manuscript PDF file)?

Reviewer #1: Yes

Reviewer #2: Yes

Reviewer #3: Yes

4. Is the manuscript presented in an intelligible fashion and written in standard English?

Reviewer #1: Yes

Reviewer #2: Yes

Reviewer #3: Yes

5. Review Comments to the Author

Reviewer #1: Thank you for the opportunity to review this paper. It is a methodological rigorous study into an important healthcare issue. Please find my questions and reviewer comments below: 

2000-2022 is a long period of time where a lot can change can occur. Additionally, there have been a lot of changes to data collection and documentation in clinical settings. Would the authors provide more insight into why this particular study window was selected?

Did the researchers conduct any statistical tests comparing changes in driving time before and after the hypothetical new clinics are added?

A little more context on what a clinic is and where it fits into the overall local health system (relative to a hospital/ large medical center) would be helpful for those who are not familiar with the Australian health system

The authors wrote in the Introduction. ““Epidemiology often focusses on the social determinants of health when describing inequity in health care delivery and outcomes. However, including a spatial perspective to these social-demographic factors can develop a more nuanced and practical model of healthcare accessibility. Tools for assessing healthcare access must measure both the geographic distribution of a patient population and their healthcare services as well as the social determinants of health that influence healthcare engagement.” This is a very important and true point. This reviewer feels that the findings from this study can be better contextualized taking into account some of the social demographic factors of the populations who are geographically underserved, rather than only discussing spatial distance. The two questions below may be considered relative to this comment. 

In the “Area Demographics Around Selected Hospital”, the authors described the IRSD scores of the selected areas for new clinics, and also described their ethnic composition. Would the authors also be able to present this information for the entire NSW area in addition to the demographic info (i.e. age, sex, etc.) presented earlier in the results? Also, would the authors be able to include a results table with the demographic and CHD information for the study population? 

Can the authors speak to if there are differences in CHD prevalence between areas with greater access to healthcare? 

Thank you for the opportunity to review this work.

Reviewer #2: The healthcare accessibility analysis in this piece is worthwhile and deserving of circulation, but the overall framing of the article in terms of the GIS itself, and it's supposed novelty, is quite puzzling actually. It seems at the very least to be named incorrectly. In terms of free-and-open-source GIS, not to say the many similar applications of enterprise health GIS, there is nothing especially novel about this approach - from a technical POV. On the one hand, there are any number of R-based Shiny and Leaflet applications showing variations on this type of healthcare access analysis, which the authors should take the time to compile as citations. On the other hand, companies like Esri have already made this type of analysis a core part of their approach to health GIS, and have extended it routinely to local health departments and many others. I believe there is even a standard "solution" template hosted through the Esri HHS department. Of course, I do take the point that Esri's solutions are not cost free, but they have clearly done quite a bit of this work already, which tends to cut into the argument that the GIS approach here is actually what is worth highlighting. 

In its current form and framing I would not recommend this piece for publication. However, if the authors reconceive their work in terms of health access analytics and emphasize the ways that a) this type of approach compares to the *many* other examples of healthcare access GIS, and b) the ways that spatial analysis itself provides novel insight into patient care, then it is perhaps worthy of reconsideration.

Reviewer #3: Hi!

The paper in actual format is looking good enough to publish, and I highly encourage this kind of paper, especially the “Data availability” concept which is nice action from the authors.

The paper has no major problems and notes however i have some comments:

The abstract is well introducing the problem and methodology, but it looks long, try to make it 200-250 words at maximum, also, I think that subtitles are unpopular in the abstract.

This paper is looking more technical paper than a thematic one, which means that the authors don’t discuss a “thematic” paper, like accessibility, distribution…etc, but focus on methods and technics, so it’s preferred to add more technical stuff in material and methods, especially technical staff about software, architecture, source code…etc.

The main issue in this paper is about using innovation technics to improve accessibility, from a geographical perspective and academic point of view, “accessibility” has not been well defined in your manuscript. Because accessibility has multiple definitions and methods to measure either, spatial, functional, acceptability..etc. I suggest adding a literature review about accessibility and related technics used in this paper,

From the formatting of this paper, I guess the article will be in good format at its final stage when publishing, however, it’s recommended to remove white spaces between paragraphs and pages

6. PLOS authors have the option to publish the peer review history of their article (what does this mean?). If published, this will include your full peer review and any attached files.

**Do you want your identity to be public for this peer review?** For information about this choice, including consent withdrawal, please see our Privacy Policy.

Reviewer #1: No

Reviewer #2: No

Reviewer #3: Yes: Belkacem LAHMAR

---

## [Decision Letter · Decision Letter 1]

11 Apr 2023

An Interactive Geographic Information System to Inform Optimal Locations for Healthcare Services

PDIG-D-22-00277R1

Dear Mr Nicholson,

We are pleased to inform you that your manuscript 'An Interactive Geographic Information System to Inform Optimal Locations for Healthcare Services' has been provisionally accepted for publication in PLOS Digital Health.

Best regards,

Yuan Lai, Ph.D.

Academic Editor

PLOS Digital Health

Reviewer Comments (if any, and for reference):

Reviewer's Responses to Questions

**Comments to the Author**

1. If the authors have adequately addressed your comments raised in a previous round of review and you feel that this manuscript is now acceptable for publication, you may indicate that here to bypass the “Comments to the Author” section, enter your conflict of interest statement in the “Confidential to Editor” section, and submit your "Accept" recommendation.

Reviewer #2: (No Response)

Reviewer #3: All comments have been addressed

2. Does this manuscript meet PLOS Digital Health’s publication criteria? Is the manuscript technically sound, and do the data support the conclusions? The manuscript must describe methodologically and ethically rigorous research with conclusions that are appropriately drawn based on the data presented.

Reviewer #2: Partly

Reviewer #3: Yes

3. Has the statistical analysis been performed appropriately and rigorously?

Reviewer #2: Yes

Reviewer #3: Yes

4. Have the authors made all data underlying the findings in their manuscript fully available (please refer to the Data Availability Statement at the start of the manuscript PDF file)?

Reviewer #2: Yes

Reviewer #3: Yes

5. Is the manuscript presented in an intelligible fashion and written in standard English?

Reviewer #2: Yes

Reviewer #3: Yes

6. Review Comments to the Author

Reviewer #2: Although the authors have added additional context in terms of the use of site selection methodologies for the sake of improving physical access to health care, it's still not entirely clear to me, apart from the specific context of CHD in NSW that qualifies this analysis as novel or in some way meaningfully advancing the field of spatial analysis for health care access.

The methods employed here are in fact routinely used by a number of health systems (for instance see Loma Linda University's work in medical GIS -- https://publichealth.llu.edu/academics/health-gis/health-gis-research) to determine efficient distributions of points of access to care. Similarly, see some of Rand's work in this area: https://www.rand.org/pubs/technical_reports/TR1146.html.

I say this not to dismiss the work outright, because clearly there is significance in the application within NSW. The authors successful make the case that site selection and patient access have been improved through the methodology described here. However, the overall framing of this paper -- in particular the titular focus on interactive GIS as a technology -- does seem to promise much more than it delivers in terms of actual technical innovation. I recommended engaging with the HealthGIS work from Esri -- such that many of these approaches are now built directly into industry standard software (https://www.esri.com/en-us/industries/health/focus-areas/access-to-care#:~:text=GIS%20allows%20you%20to%20match,and%20inform%20and%20educate%20patients.) It would be a good idea to think through exactly how the open source approach here relates to other GIS industry approaches. Likewise, it's not especially clear, apart from economic considerations around the cost of analytical systems implementations (which are mostly unexplored here) what the value is to the argument of focusing on the ability to do this sort of analysis in R-Shiny.

Honestly, I think this paper would have been better off reframing around the issues and findings in NSW rather than on the technology itself, just given that the use of the technology is mostly reiterating methodological work that is by now fairly well defined.

Reviewer #3: REVIEWING An Interactive Geographic Information System to Inform Optimal Locations for Healthcare Services ON PLOS DIGITAL HEALTH REV 1

Dear authors:

I would like to express my gratitude for the method that use to respond to reviewer comments, the table was very good, and I will use your method in my future works.

About the manuscript

The authors take into consideration the previous note about minimising the abstract and removing subtitles, now it looks better, however, in the future try to follow a more straightforward approach which means introducing the problem, after that your approach briefly (no details) and your contributions, finally the outputs and result. Please don’t change your manuscript abstract this time because it's looking good.

The adjusted all notes mentioned before by reviewers (as I see), and he adds more technical details about the software and methods (Opensource are used widely), The data is available online in addition to the application.

I think this manuscript is well enough to accept to pulpish in your journal. I have no furthermore comments. Waiting for other reviewers' decision.

Good luck

7. PLOS authors have the option to publish the peer review history of their article (what does this mean?). If published, this will include your full peer review and any attached files.

**Do you want your identity to be public for this peer review?** For information about this choice, including consent withdrawal, please see our Privacy Policy.

Reviewer #2: **Yes: **Andrew Schroeder

Reviewer #3: **Yes: **Belkacem Lahmar
